# Higher spin partition functions via the quasinormal mode method in de Sitter quantum gravity

**Victoria L. Martin$^\star$ and Andrew Svesko**

Department of Physics, Arizona State University, Tempe, Arizona 85287, USA

$\star$ victoria.martin.2@asu.edu

## Abstract

In this note we compute the 1-loop partition function of spin-$s$ fields on Euclidean de Sitter space $S^{2n+1}$ using the quasinormal mode method. Instead of computing the quasinormal mode frequencies from scratch, we use the analytic continuation prescription $L_{\text{AdS}} \to i L_{\text{dS}}$, appearing in the dS/CFT correspondence, and Wick rotate the normal mode frequencies of fields on thermal AdS$_{2n+1}$ into the quasinormal mode frequencies of fields on de Sitter space. We compare the quasinormal mode and heat kernel methods of calculating 1-loop determinants, finding exact agreement, and furthermore explicitly relate these methods via a sum over the conformal dimension. We discuss how the Wick rotation of normal modes on thermal AdS$_{2n+1}$ can be generalized to calculating 1-loop partition functions on the thermal spherical quotients $S^{2n+1}/\mathbb{Z}_p$. We further show that the quasinormal mode frequencies encode the group theoretic structure of the spherical spacetimes in question, analogous to the analysis made for thermal AdS in [1–3].

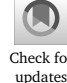

# 1 Introduction

In de Sitter quantum gravity the chief object of interest is the Euclidean partition function,

$$Z = \int \mathcal{D}g \, e^{-S[g]} \, , \tag{1}$$

written here as a path integral over all compact Euclidean metrics $g$. The partition function (1) is interpreted as the norm of the Hartle-Hawking state, i.e., the vacuum state wavefunctional with fixed boundary conditions on a space-like slice [4].

Formally, the full partition function (1) is computed using a saddle point approximation,

$$Z = \sum_{g_c} e^{-\alpha S^{(0)}[g_c] + S^{(1)}[g_c] + \frac{1}{\alpha} S^{(2)}[g_c] + \cdots} \, , \tag{2}$$

where $g_c$ are classical solutions to the Euclidean equations of motion, and $\alpha$ is a dimensionless coupling constant equal to the de Sitter radius in Planck units. In the exponential $S^{(i)}[g_c]$ represents the $i$th quantum correction to the Euclidean action $S$; $S^{(0)}[g_c]$ is the tree-level contribution, $S^{(1)}[g_c]$ is the 1-loop quantum correction providing information on leading order quantum effects, and so forth.

Computing the full partition function (1) boils down to identifying the infinite set of classical solutions $g_c$, and evaluating the infinite series of contributions $S^{(i)}$ about each solution $g_c$. The first step for de Sitter gravity is straightforward: the infinite set of classical solutions is enumerated by Euclidean dS$_N$, i.e., the sphere $S^N$, and its quotients $S^N/\Gamma$, for $\Gamma$ a discrete subgroup of $SO(N+1)$. The more difficult part of the procedure is carrying out the saddle point approximation (2) in full[1].

The partition function is therefore typically computed to 1-loop. The leading order quantum correction is given by the 1-loop partition function $Z^{(1)}$ and is given in terms of a collection of functional determinants of kinetic operators. For example, in $D$-dimensional (Euclidean) de Sitter space $Z^{(1)}$ is given as a ratio of two functional determinants [8,9]

$$Z^{(1)}_{\text{grav}} = \frac{\sqrt{\det\left(\nabla_1^2 - \frac{2R}{D}\right)}}{\sqrt{\det\left(\nabla_2^2 - \frac{2R}{D}\right)}} \, . \tag{3}$$

The denominator comes from linearized metric fluctuations while the numerator represents the contribution of a spin-1 ghost originating from the Fadeev-Popov determinant due to a gauge fixing condition. Here $\nabla_2^2$ and $\nabla_1^2$ represent the (Lichnerowicz) Laplacians acting on a rank-2 tensor and spin-1 vector, respectively, and $R$ is the Ricci scalar.

Generally then, we are interested in computing the functional determinants of kinetic operators of massive spin-$s$ fields on the sphere and its discrete quotients. There exist multiple ways of computing 1-loop determinants. One approach, the heat kernel method, involves calculating the heat kernel between two spacetime points $K(x, y; t)$ on the given space. The 1-loop partition function is then related to the Mellin transformation of the trace of the coincident heat kernel. This was carried out explicitly for $S^N$ and the Lens space quotients $S^N/\mathbb{Z}_p$ in [10, 11]. The basic idea of [10, 11] utilizes the fact that the sphere and its quotients are homogeneous spaces, and builds the heat kernel on $S^N$ using group theoretic techniques developed in [12–14]. The 1-loop partition function on $S^N/\Gamma$ is then computed using the method

---

[1]There are special cases, however, where the full partition function can be computed exactly, e.g., quantum general relativity and topologically massive gravity in three-dimensional de Sitter space, dS$_3$ [5,6], where the infinite series of perturbative corrections is evaluted using the relationship between three-dimensional Einstein gravity and Chern-Simons theory [7]

of images,

$$K^{S^N/\Gamma}(x, y; t) = \sum_{\gamma \in \Gamma} K^{S^N}(x, \gamma(y); t), \tag{4}$$

where $\gamma$ are the generators of $\Gamma$. This method was used to explicitly compute 1-loop partition functions of symmetric, transverse, traceless (STT) tensors on $S^3$ and Lens spaces $S^3/\mathbb{Z}_p$ in [10], and the STT heat kernel for higher odd-dimensional spheres $S^{2n+1}$ and Lens spaces $S^{2n+1}/\mathbb{Z}_p$ in [11].

Another way to compute 1-loop partition functions, known as the the quasinormal mode method [15, 16], assumes $Z^{(1)}(\Delta)$ is a meromorphic function in a mass parameter $\Delta$ and utilizes the Weierstrass factorization theorem,

$$Z^{(1)}(\Delta) = e^{\text{Poly}(\Delta)} \frac{\prod_{\Delta_0} (\Delta - \Delta_0)^{d_0}}{\prod_{\Delta_p} (\Delta - \Delta_p)^{d_p}}. \tag{5}$$

Here we have decomposed $Z^{(1)}(\Delta)$ into a product of its poles $\Delta_p$ with degeneracy $d_p$ and zeros $\Delta_0$ of degeneracy $d_0$, up to an entire function $\text{Poly}(\Delta)$. For cases of interest to us, $Z^{(1)}(\Delta)$ will only have poles[2].

The key insight of [15, 16] is that, for fields on backgrounds with horizons, the poles $\Delta_p$ occur at a discrete set of values, $\omega_*(\Delta)$, identified as the (Wick rotated) quasinormal modes[3]. Here $*$ represents a collection of quantum numbers, such as angular momentum. For example, $Z^{(1)}(\Delta)$ for a massive scalar field $\phi$ living on thermal backgrounds at temperature $T$ is

$$Z^{(1)}(\Delta) = e^{\text{Poly}(\Delta)} \prod_{n,*} (\omega_n(T) - \omega_*(\Delta))^{-1}, \tag{6}$$

where $\omega_n(T)$ are the Matsubara frequencies arising from a periodicity condition of the Euclidean time direction imposed on the field $\phi$. Thermal spacetimes that define a canonical ensemble, e.g., the sphere $S^N$, have $\omega_n(T) = 2\pi i n T$, while spacetimes defining a grand canonical ensemble, such as the Lens spaces $S^N/\mathbb{Z}_p$, will have Matsubara frequencies which depend on the temperature and angular potentials. The quasinormal mode method was used to compute 1-loop partition functions of scalar fields on AdS-Schwarzschild black holes, Euclidean $dS_{d+1}$ and thermal $AdS_3$ in [15], arbitrary spin fields on a non-rotating BTZ black hole in [18, 19], and the graviton on a rotating BTZ black hole in [20]. Thus far the quasinormal method has not been used to compute 1-loop partition functions on quotients of the sphere.

In this note we aim to compute 1-loop partition functions of spin-$s$ fields on $S^{2n+1}$ and its associated Lens spaces $S^{2n+1}/\mathbb{Z}_p$ using the quasinormal mode method, and see how it compares to the heat kernel method. Lens spaces are of interest when studying quantum corrections, as we further review in Section 2. Rather than computing the quasinormal mode frequencies for spin-$s$ fields outright, we will take advantage of the fact that quasinormal frequencies for scalar fields on Euclidean de Sitter space Wick rotate to normal frequencies for a scalar field on thermal AdS [15, 21]. Using this observation we utilize recent work studying the normal mode frequencies of spin-$s$ fields on thermal $AdS_{2n+1}$ [3] to compute the associated 1-loop partition functions. We find perfect agreement with the heat kernel method, where we explicitly show that a sum over the mass parameter $\Delta$ carries all of the information of the eigenvalues of the

---

[2]Here we will consider functional determinants of higher spin fields. Note that while the partiton function may have zeros, *e.g.*, (3), the functional determinants may be expanded by their poles. This is because, for example, (3) breaks down into a product of functional determinants of massive fields; the massless graviton is constructed from the partition functions of massive spin-2 tensor, a massive spin-1 vector, and massive scalar functional determinant, see, *e.g.*, Eqn. (4.21) of [17].

[3]Quasinormal modes are eigenmodes of dissipative systems, such as those modes obeying infalling boundary conditions at a black hole event horizon, or the cosmological horizon of de Sitter space.

Laplacian $\nabla^2_{(s)}$ and the sums over quasinormal mode frequency integers build the degeneracy formula of the eigenvalues. Therefore, we show that the quasinormal modes encode the group theoretic structure of the spaces the fields live on, analogous to the analysis made for thermal AdS [1–3].

The outline of this note is as follows. Section 2 is devoted to a brief review the geometry and physics of Euclidean de Sitter space in odd dimensions and its Lens space quotients. We then build the 1-loop partition function for higher spin fields living on the sphere and its thermal quotients using the quasinormal mode method in Section 3. We summarize our findings and discuss potential future work in Section 4.

## 2 Field theory on Euclidean dS$_{2n+1}$ and its thermal quotients

The physics of timelike observers in de Sitter space dS$_D$ is best described using static patch coordinates

$$\frac{ds^2}{L^2} = -\cos^2\theta \, dt^2 + d\theta^2 + \sin^2\theta \, d\Omega^2_{D-2} \,, \tag{7}$$

for $0 < \theta < \pi/2$, and where $L$ is the radius of curvature. The metric manifestly has a timelike Killing vector, an isometry inherited from Minkowski space preserving the de Sitter hyperboloid. Observers are only in causal contact with a portion of the full de Sitter geometry called the causal diamond, with a boundary at the cosmological horizon located at $\theta = \pi$.

Studying quantum field theory in de Sitter space requires choosing a vacuum state. The common choice is the Hartle-Hawking state defined by the analytic continuation from Euclidean signature. Under the Wick rotation $t_E = it$, we obtain Euclidean de Sitter space in static patch coordinates

$$\frac{ds_E^2}{L^2} = \cos^2\theta \, dt_E^2 + d\theta^2 + \sin^2\theta \, d\Omega^2_{D-2} \,. \tag{8}$$

In order for the geometry to be non-singular at $\theta = \pi/2$ we require that the Euclidean time coordinate $t_E$ be periodically identified, i.e., $t_E \sim t_E + 2\pi m$ for $m \in \mathbb{Z}$. We also recognize (8) as the metric of $S^D$. For example, Euclidean dS$_3$ in static coordinates is the metric of $S^3$ in Hopf coordinates

$$ds_{S^3}^2 = d\theta^2 + \cos^2\theta \, dt_E^2 + \sin^2\theta \, d\phi^2 \,, \tag{9}$$

with the identifications

$$(t_E, \phi) \sim (t_E + 2\pi m, \phi + 2\pi n) \,, \tag{10}$$

for $n, m \in \mathbb{Z}$. Euclidean dS$_5$ is just $S^5$

$$ds_{S^5}^2 = d\theta^2 + \cos^2\theta \, dt_E^2 + \sin^2\theta \, ds_{S^3}^2 \,, \tag{11}$$

built from the six real coordinates $\{x_i\}$ satisfying $x_i^2 = 1$

$$
\begin{aligned}
x_1 &= \cos\theta \cos t_E \,, \quad x_2 = \cos\theta \sin t_E \,, \quad x_3 = \sin\theta \cos\psi \cos\phi_1 \,, \\
x_4 &= \sin\theta \cos\psi \sin\phi_1 \,, \quad x_5 = \sin\theta \sin\psi \cos\phi_2 \,, \quad x_6 = \sin\theta \sin\psi \sin\phi_2 \,.
\end{aligned}
\tag{12}
$$

The identifications

$$(t_E, \phi_2, \phi_3) \sim (t_E + 2\pi m, \, \phi_1 + 2\pi n_1, \, \phi_2 + 2\pi n_2) \,, \quad m, n_1, n_2 \in \mathbb{Z} \tag{13}$$

ensure the geometry (11) is non-singular at the horizon $\theta = \pi/2$.

In general, Euclidean $dS_{2n+1}$ is $S^{2n+1}$

$$ds^2 = d\theta^2 + \cos^2\theta\, d\phi_1^2 + \sin^2\theta\, d\Omega_{2n-1}^2 \,, \tag{14}$$

where in order for the geometry to be regular at $\theta = \pi/2$, we require each of the phases[4] $(t_E, \phi_1, ..., \phi_n)$ be periodicially identified

$$(t_E, \phi_1, ..., \phi_n) \sim (t_E + 2\pi m,\ \phi_1 + 2\pi n_1,\ ...,\ \phi_n + 2\pi n_n)\,, \quad m, n_1, n_2, ..., n_n \in \mathbb{Z}\,. \tag{15}$$

Observers living in the static patch compute field theory correlators with respect to the Hartle-Hawking vacuum state. The quantum state can be defined in terms of a density matrix

$$\rho = e^{-\beta H}\,, \tag{16}$$

where $\beta = 2\pi L$, and $H = i\partial_t$ is the Hamiltonian. The state (16) is also the generator of the identification (10). Physically the state $\rho$ is precisely a thermal density matrix of a canonical ensemble at a fixed temperature $\beta^{-1} = T = 1/2\pi L$. That is, an observer in Euclidean de Sitter space sitting at $\theta = 0$ will detect thermal radiation originating from the cosmological horizon at temperature $T = 1/2\pi L$.

There are additional identifications to the coordinates $(t_E, \phi_1, ..., \phi_n)$ which also make the Euclidean geometry (15) smooth. Specifically, given the set of $n$ integers $q_i$ coprime with an integer $p$ (i.e. $\gcd(p, q_i) = 1$, the following set of identifications also make (15) smooth:

$$(t_E, \phi_1, ..., \phi_n) \sim \left(t_E + 2\pi\frac{m}{p}, \phi_1 + 2\pi n_1 + 2\pi m\frac{q_1}{p}, ..., \phi_n + 2\pi n_n + 2\pi m\frac{q_n}{p}\right)\,. \tag{17}$$

These identifications define the Lens space $L(p, q_1, ..., q_n)$, which is the 'thermal' quotient of the sphere $S^{2n+1}$ by the cyclic group of order $p$, $\mathbb{Z}_p$, i.e., $L(p, q_1, ..., q_n) \equiv L(p, q_i) = S^{2n+1}/\mathbb{Z}_p$. For example, $S^3/\mathbb{Z}_p = L(p, q)$, with $L(1, 0) = S^3$. The integers $q_i$ label the various ways of embedding $\mathbb{Z}_p$ into the isometry group $SO(2n+2)$ of $S^{2n+1}$. Note that the shift $q_i \to q_i + \ell_i p$ for integer $\ell_i$ can be absorbed into a change of integers $m, n_i$, and therefore $q_i$ is defined mod $p$.

Quantum field theory on $L(p, q_i)$, in light of the identifications (17), has a straightforward physical interpretation: a Lens space describes a grand canonical ensemble, with a density matrix

$$\rho = e^{-\frac{2\pi L}{p}(H + iq_1 J_1 + iq_2 J_2 + ... + iq_n J_n)}\,, \tag{18}$$

where $J_i = i\partial_{\phi_i}$ is the axial symmetric Killing vector associated with each $\phi_i$, representing an angular momentum. To an observer in the static patch, the Lens space provides a means of preparing the quantum state, where here they would detect a finite temperature $\beta^{-1} = p/2\pi L$ bath of radiation with fixed angular potentials $\vartheta_i = 2\pi Liq_i/p$ emitted from the cosmological horizon at $\theta = \pi/2$.

Lens spaces represent genuine quantum gravitational effects. This is because because in the saddle point approximation (2) each Euclidean geometry $g_c$ is weighted by minus its action, which is proportional to the volume of $g_c$. Since Lens spaces all have a lower volume than $S^{2n+1}$, the contribution from the $S^{2n+1}$ saddle dominates the sum in the limit the de Sitter radius is large, $L/G \gg 1$. Therefore, the Lens spaces are suppressed by terms that are exponentially large in the de Sitter entropy in the sum over geometries, and their contributions

---

[4]Here we define coordinates on $S^{2n+1}$ in terms of complex numbers $(z_E, ..., z_n)$, carrying phases $t_E, \phi_1, ..., \phi_n$, respectively, and satisfying $|z_1|^2 + ... + |z_{n+1}|^2 = 1$. The $n+1$ complex numbers $z_i$ can be further decomposed into $2n+2$ real cordinates $x_i$, as shown for $S^5$. For $S^{2n}$ it is easier to work with the ordinary spherical coordinate system.

lead to deviations from the standard thermality of Euclidean de Sitter space[5].

We will suggestively introduce parameters

$$\tau_i \equiv \vartheta_i - \beta = \frac{2\pi L q_i}{p} - \frac{2\pi L}{p} \,, \quad \bar{\tau}_i \equiv \vartheta_i + \beta = \frac{2\pi L q_i}{p} + \frac{2\pi L}{p} \,, \tag{19}$$

reminding us of the modular parameters $\tau_i = \vartheta_i + i\beta$ of hyperbolic space $H^{2n+1}/\mathbb{Z}$, i.e., thermal $AdS_{2n+1}$ at temperature $\beta^{-1}$ with angular potentials $\vartheta_i$. There are important differences between the modular parameters in $dS_{2n+1}$ and $AdS_{2n+1}$, leading to different physics. The Lens space $L(p, q_i)$ cause the modular parameters of $dS_{2n+1}$ to only take distinct rational values, while the modular parameter of thermal $AdS_{2n+1}$ is a continuous complex variable. Physically these differences correspond to the fact that there are only a discrete set of values of temperature and angular potential describe static patches of $dS_{2n+1}$, while the temperature and angular potentials in thermal $AdS_{2n+1}$ take on continuous values

So far we have only considered the thermal quotients $S^{2n+1}/\Gamma$ when $\Gamma$ is the abelian group $\mathbb{Z}_p$. There are additional discrete, freely acting subgroups of $SO(2n + 2)$ which should be considered when computing the full partition function. For example, for $S^3/\Gamma$ the additional possible $\Gamma$ are non-abelian and are central extensions of crystallographic groups, e.g., the central extension of the dihedral group. A complete classification is given in [23]. The contributions of these additional quotients were incorporated into the tree level partition function in [5], however, it was shown that the classical partition function will continue to diverge even after zeta function regularization. For this reason we won't focus on these quotients of the sphere. The functional determinants of the Lens spaces and the dihedral case have been worked out and regularized using the Barnes zeta function in [24–26].

# 3   1-Loop Partition functions

## 3.1   Quasinormal Mode Method

**Euclidean de Sitter**

Let us now compute the 1-loop partition functions for arbitrary spin-$s$ fields on Euclidean $dS_{2n+1}$ and its thermal quotients $S^{2n+1}/\mathbb{Z}_p$, focusing particularly on the three-dimensional case. This can be done using the heat kernel method [10, 11] and was used to compute the 1-loop graviton partition function on $S^3$ and $S^3/\mathbb{Z}_p$ in [5]. Instead we will compute the 1-loop partition function using the quasinormal mode method, which applies equally well to thermal de Sitter space, and show explicitly how the quasinormal modes construct the heat kernel. The calculation for the 1-loop partition function for scalar fields on $S^{d+1}$ was carried out in [15], which we will briefly review before we generalize to spin-$s$ fields on $S^{2n+1}$ and $S^{2n+1}/\mathbb{Z}_p$.

The quasinormal method of computing the 1-loop partition function $Z^{(1)}$ assumes that the 1-loop partition function is a meromorphic function in a mass parameter $\Delta$, allowing us to use the Weierstrass factorization theorem (5). Since the $Z^{(1)}$ for scalar fields has no zeros, we write

$$Z^{(1)}(\Delta) = \text{Poly}(\Delta) \prod_{\Delta_p} (\Delta - \Delta_p)^{-d_p} \,, \tag{20}$$

---

[5]Three dimensional Lens spaces $L(p, q)$ are particularly interesting in mathematics and physics. Topologically $L(p, q)$ can be pictured as two solid tori glued together along their $T^2$ boundaries, where each $L(p, q)$ corresponds to a different gluing. A simple gluing maps the contractible cycle in one torus to the dual non-contractible cycle of the other torus, leading to the sphere $S^3 = L(1, 0)$. More complicated gluings resulting in more complicated Lens spaces exist, some of which are described in [5], where they showed the sum over such Lens spaces can be regarded as a sum over possible cycles that are contractible at the cosmological horizon, which can be interpreted as the de Sitter equivalent of the black hole Farey tail [22].

where $\Delta_p$ are the poles of $Z^{(1)}(\Delta)$, each with degeneracy $d_p$, and occur when $\det(-\nabla^2 + m_0^2) = 0$. These will occur whenever there exists a smooth scalar field solution to the Klein Gordon equation in $d + 1$-dimensional spacetime.

$$\left(-\nabla^2 + \frac{\Delta_p(\Delta_p - d)}{L^2}\right)\phi_p = 0, \quad \Delta(\Delta - d) = -m_0^2, \tag{21}$$

on the Euclidean continuation of dS, satisfying periodic boundary conditions in Euclidean time, which we will also refer to as thermal boundary conditions. The so-called Euclidean zero modes $\phi_p$ occur at complex $\Delta_p$; for compact spaces such as the sphere these $\Delta_p$ can be labeled by a discrete index.

For scalar fields on $S^{d+1}$, to demand $\phi_p$ satisfy thermal boundary conditions we require $\Delta = \Delta_p$, for

$$-\Delta_p \geq \ell \geq m_{d-1} \geq m_2 \geq |m_1|, \tag{22}$$

where $\ell, m_i$ are the quantum numbers for spherical harmonics in $S^{d+1}$, and $\Delta_p \in \mathbb{Z}$. Moreover, at each non-positive integer $\Delta_p \equiv -r$ we have a Euclidean zero mode with degeneracy [12]

$$d(r) = \frac{2r + d}{d}\binom{r + d - 1}{d - 1}. \tag{23}$$

For example, for $d = 1$, $d(r) = 2r + 1$, for $d = 2$, $d(r) = (r+1)^2$, and so forth. Following [15] the 1-loop partition function for a scalar field (20) is given by[6]

$$\log Z^{(1)}(\Delta_{\pm}) - \text{Poly}(\Delta_{\pm}) = -\sum_{\pm}\sum_{r=0}^{\infty} d(r)\log(r + \Delta_{\pm}), \tag{24}$$

with

$$\Delta_{\pm} = \frac{d}{2} \pm \sqrt{\left(\frac{d}{2}\right)^2 - m_0^2 L^2}, \tag{25}$$

where $m_0$ is the mass of the scalar field. We will give a physical interpretation of $\Delta_{\pm}$ momentarily.

The entire function $\text{Poly}(\Delta_{\pm})$ encodes the UV divergences of the problem. This contribution can be determined by studying the large $\Delta_{\pm}$ behavior of $Z^{(1)}(\Delta_{\pm})$ and matching to the heat kernel curvature expansion of $Z^{(1)}(\Delta)$. In other words, $\text{Poly}(\Delta_{\pm})$ is fixed by requiring $\log Z^{(1)}(\Delta_{\pm})$ found using quasinormal mode methods match the 1-loop partition function found using a heat kernel coefficient expansion in the large $\Delta$ limit. As shown in [15], $\text{Poly}(\Delta)$ for scalar fields living on odd-dimensional spheres vanishes exactly. As we will see later, upon comparing to heat kernel methods, we likewise observe $\text{Poly}(\Delta_{\pm}) = 0$ for higher spin-$s$ fields in the odd-dimensional spheres we consider. Moreover, as we will see shortly, $\text{Poly}(\Delta_{\pm})$ does not carry any temperature dependent information[7], and we will therefore not be interested in this contribution when we compare the heat kernel and quasinormal mode methods. From here on, we will drop any reference to $\text{Poly}(\Delta_{\pm})$.

Alternatively, it was recognized in [15] that the Euclidean zero modes $\phi_p$ are identified as the Wick rotated Lorentzian quasinormal frequencies for a scalar field on $dS_{d+1}$ [27]

$$\omega_{\tilde{p},\ell} = \pm 2\pi i T(2\tilde{p} + \ell + \Delta_{\pm}). \tag{26}$$

---

[6]Here we explicitly included the $\Delta_{\pm}$ roots of $\Delta(\Delta - d) = -m_0^2 L^2$, such that for $m_0 L > d/2$, $\Delta_{\pm}$ are complex. For the sphere $S^{d+1}$ we must include contributions from both $\Delta_{\pm}$ because there is no boundary, contrary to the case of thermal $AdS_{d+1}$, where the boundary conditions fix the root $\Delta$.

[7]This is made manifest if we choose not to set $T = \frac{1}{2\pi}$.

Here $\tilde{p}$ and $\ell$ are the radial and angular momentum quantum numbers, respectively, where $\tilde{p}, \ell = 0, 1, 2, ..., T$ is the temperature of Euclidean de Sitter, and $\Delta_\pm$ is given in (25).

The condition that $\phi_p$ satisfy periodic Euclidean time boundary conditions transforms into the quasinormal frequencies (26) being set to the thermal Matsubara frequencies $\omega_n = 2\pi i n T$, where $T = 1/2\pi L$ is the temperature of Euclidean dS, and $n \in \mathbb{Z}$ is a thermal integer. The sign of $n$ encodes what type of boundary conditions the solutions to the Klein-Gordon equation satisfy: for $n > 0$ the Euclidean zero modes $\phi_p$ satisfy ingoing boundary conditions at the horizon, whose real time Wick rotation are the quasinormal modes (26); for $n < 0$ $\phi_p$ satisfies outgoing boundary conditions, which Wick rotate to anti-quasinormal modes, and $\bar{\omega}_{\tilde{p},\ell} = \omega^*_{\tilde{p},\ell}$, where $*$ denotes complex conjugation. The $n = 0$ Euclidean zero modes can be treated either as a quasinormal or antiquasinormal frequencies.

Altogether, the product over the Euclidean zero modes in the 1-loop partition function (20) translates to a product over the quasinormal frequencies, where the poles occur when $\omega_{\tilde{p},\ell} = \omega_n$. Therefore, the 1-loop partition function is computed using (6). Explicitly, for a scalar field on $S^3$,

$$
\begin{aligned}
\log Z^{(1)}(\Delta_\pm) &= \log \prod_\pm \prod_{\tilde{p},\ell} \sqrt{(\omega_0 - \omega_{\tilde{p},\ell})(\omega_0 - \bar{\omega}_{\tilde{p},\ell})} \prod_{n>0} (\omega_n - \omega_{\tilde{p},\ell})^{-1} \prod_{n<0} (\omega_n - \bar{\omega}_{\tilde{p},\ell})^{-1} \\
&= \sum_\pm \sum_{\tilde{p},\ell=0}^\infty \left[ \log(2\tilde{p} + \ell + \Delta_\pm) - 2 \sum_{n=0}^\infty \log(n + 2\tilde{p} + \ell + \Delta_\pm) \right] \\
&= -\sum_\pm \sum_{k=0}^\infty (k+1) \left[ \log(k + \Delta_\pm) - 2 \sum_{n=0}^\infty \log(n + k + \Delta_\pm) \right] \\
&= -\sum_\pm \sum_{r=0}^\infty (r+1)^2 \log(r + \Delta_\pm) .
\end{aligned}
\tag{27}
$$

To get to the third line we combined the sums over $\tilde{p}, \ell$ with degeneracy[8] of $S^1$ into a single sum over $k$ with degeneracy $k+1$, and to get to the final line we perform the sum over $k$ and $n$ and write the resulting expession in a more compact form. We recognize that equation (27) matches the 1-loop partition function in (24), where we see that the sum over quasinormal mode quantum numbers $p$ and $\ell$ build the degeneracy formula for a scalar field with total angular momentum $r$ on $S^3$, $d_r = (r+1)^2$, matching (24). The above calculation can be readily generalized to scalar fields on $S^{d+1}$ by simply noting that the only change is the degeneracy in angular momentum quantum number $\ell$ is $S^{d-1}$.

The appearance of the mass parameter (25) is suggestive of something fundamental: there is a conformal field theory (CFT) dual to de Sitter space, as in dS/CFT [28–30], and $\Delta_\pm$ represents the conformal dimension of the dual CFT operator to a scalar field. While the quasinormal modes (26) were derived independent of any CFT data, the mass function which appears naturally when computing the 1-loop partition function $Z^{(1)}(\Delta)$ is the conformal dimension (25). In fact, the quasinormal modes (26) are the Wick rotated *normal* modes of a scalar field in thermal $\text{AdS}_{d+1}$[9] via the analytic continuation [30]

$$
L_{\text{AdS}} = i L_{\text{dS}} .
\tag{28}
$$

---

[8]The degeneracy $d_\ell$ of each $d + 1$-dimensional quasinormal frequency equals the degeneracy of the $\ell$th angular momentum eigenvalue of $S^{d-1}$; for $S^3$, the degeneracy in $\omega_{\tilde{p},\ell}$ is the degeneracy of the angular momentum eigenvalue for $S^1$, i.e., $d_0 = 1$, and $d_\ell = 2$ for $\ell > 0$. As a result the variable $k \equiv 2\tilde{p} + \ell$ has degeneracy $k+1$.

[9]Thermal $\text{AdS}_{d+1}$ is Euclidean $\text{AdS}_{d+1}$ with its Euclidean time coordinate $t_E$ periodically identified with $t_E \sim t_E + 2\pi\beta$, where $\beta$ is the inverse temperature. Topologically, thermal $\text{AdS}_{d+1}$ is the quotient of the upper half plane, $\mathbb{H}^{d+1}/\mathbb{Z}$, described by a solid torus with modular parameters $\tau_i = \theta_i + i\beta$, with $\theta$ being an angular potential. The normal modes for a scalar field are $\omega^{\text{AdS}} = -(2p + \ell + \Delta)$, where $\Delta$ is the Wick rotated conformal dimension of a CFT operator dual to a scalar field.

This is the same Wick rotation which takes correlation functions on Euclidean AdS into correlation functions on Lorentzian de Sitter. This provides suggestive evidence supporting the dS/CFT conjecture, relating the central charges of the CFTs living at the boundary of AdS or de Sitter, $c_{\text{AdS}} \to i c_{dS}$, consistent with the dS/CFT conjecture [30].

In some ways the partition function of Euclidean dS can be understood as the analytic continuation of the partition function of Euclidean AdS[10]. From the heat kernel method perspective this is easy to understand mathematically, as the heat kernel of the sphere and its thermal quotients are the double Wick rotations of the heat kernel of Euclidean AdS and its thermal quotients [10, 11]. From the viewpoint of the quasinormal mode method, however, the Wick rotation that translates between the two problems is coming from Maldacena's analytic continuation prescription (28).

We will take advantage of the above observation to work out the 1-loop partition function for higher spin fields on Euclidean $dS_{2n+1}$. For example, the analytically continued normal modes for spin-$s$ fields on a thermal $AdS_{2n+1}$ [3, 18, 19] background lead to the quasinormal modes for a spin-$s$ field on $dS_{2n+1}$:

$$\omega_{\tilde{p},\ell,s} = \pm \frac{i}{L}(2\tilde{p} + \ell + \Delta_{\pm} \pm s)\,, \tag{29}$$

where now

$$\Delta_{\pm} = n \pm \sqrt{s + n^2 - m_s^2 L^2}. \tag{30}$$

When we compute the 1-loop partition function for a spin-$s$ field on $S^{2n+1}$, we start from (6), this time with frequencies (29), where the only real change to (27) is the form of the degeneracy formula (23)

$$\log Z^{(1)}(\Delta_{\pm}) = -\sum_{\pm}\sum_{r=0}^{\infty} d^{(s)}(r)\log(r + s + \Delta_{\pm})\,, \tag{31}$$

where

$$d^{(s)}(r) = (2 - \delta_{s,0})\prod_{i=1, i<j}^{n+1}\prod_{j=1}^{n+1}\frac{\ell_i^2 - \ell_j^2}{\mu_i^2 - \mu_j^2}\,, \quad \mu_i = (n+1) - i\,, \quad \ell_i = m_i + (n+1) - i\,. \tag{32}$$

Here $m_i = (m_1, m_2, ..., m_{n+1})$ characterizes the unitary irreducible representations of $SO(2n+2)$. For STT tensors $m_1 = r + s, m_2 = s$, and all other $m_{i>2} = 0$ [11]. When $d = 2n + 1$, and $s = 0$ we recover the degeneracy formula for scalar fields with total angular momentum $r$ on $S^{2n+1}$, (23). When $n = 1$ we have the degeneracy for spin-$s$ fields on $S^3$ [10]

$$d_r^{(s)} = (2 - \delta_{s,0})\left(\frac{\ell_1^2 - \ell_2^2}{\mu_1^2 - \mu_2^2}\right) = (2 - \delta_{s,0})[(r + s + 1)^2 - s^2]\,. \tag{33}$$

For $n = 2$ (32) gives us the degeneracy for STT tensors on $S^5$

$$\begin{aligned} d_r^{(s)} &= (2 - \delta_{s,0})\left(\frac{\ell_1^2 - \ell_2^2}{\mu_1^2 - \mu_2^2}\right)\left[\left(\frac{\ell_1^2 - \ell_3^2}{\mu_1^2 - \mu_3^2}\right)\left(\frac{\ell_2^2 - \ell_3^2}{\mu_2^2 - \mu_3^2}\right)\right] \\ &= \frac{(2 - \delta_{s,0})}{2 \cdot 3!}(s+1)^2(r+s+2)^2\left[(r+s+2)^2 - (s+1)^2\right]\,. \end{aligned} \tag{34}$$

---

[10]In our analysis we are summing over specific spaces, namely the sphere and its Lens space quotients. In the full partition function one performs a sum over all types of spaces that contribute to the Euclidean path integral.

We can regulate the sum in (31) using Hurwitz zeta function[11] technology, as shown in [15]. From the definition of the Hurwitz zeta function we can write in general that for a polynomial $Q(r)$,

$$-\sum_{r=0}^{\infty} Q(r)\log(r+x) = [Q(-x+\delta_z)\zeta'(z,x)]_{z=0} \,, \tag{35}$$

where $\delta_z f(z) \equiv f(z-1)$. Therefore, explicitly for spin-$s$ fields on $S^3$, with $x \equiv s + \Delta_\pm$,

$$\log Z^{(1)}(\Delta_\pm) = \sum_\pm [d^{(s)}(-x+\delta_z)\zeta'(z,x)]_{z=0}$$

$$= (2-\delta_{s,0})\sum_\pm \Big(\zeta'(-2,s+\Delta_\pm) + 2(1-\Delta_\pm)\zeta'(-1,s+\Delta_\pm) \tag{36}$$

$$+ (\Delta_\pm - 1 + s)(\Delta_\pm - 1 - s)\zeta'(0,s+\Delta_\pm)\Big).$$

When $s=0$, we recover the scalar field result for $S^3$ found in [15]. We can similarly workout the expression for higher spin fields. For example, for the massless graviton we evaluate

$$\log Z^{(1)}_{S^3,\mathrm{grav}} = -\frac{1}{2}\log\det[-\nabla^2_{(2)} + 2] + \frac{1}{2}\log\det[-\nabla^2_{(1)} - 2]\,. \tag{37}$$

Using the conformal dimensions associated with a massive spin-2 and spin-1 field, $\Delta^{(2)}_\pm = 1\pm 1$ and $\Delta^{(1)}_\pm = 1 \pm 2$ (see Equation (30)), we find after some algebra and using $\zeta'(0,x) = \log(\Gamma(x)) - \frac{1}{2}\log(2\pi)$,

$$\log Z^{(1)}_{S^3,\mathrm{grav}} = \log[(2\pi)^6] - \log(2^8)\,, \tag{38}$$

or

$$Z^{(1)}_{S^3,\mathrm{grav}} = \frac{\pi^6}{4}\,. \tag{39}$$

This matches what was found using the heat kernel method in [5][12].

Before we move to comment on the quasinormal mode analysis for Lens space quotients, let us briefly point out another method of computing the 1-loop partition function for scalar fields on $S^{d+1}$ [24, 25, 31]. This is accomplished introducing additional radial and angular momenta quantum numbers $\tilde{p}_i$ and $\ell_i$, respectively, where each $\ell_i$ has the degeneracy of the angular momentum eigenvalue for $S^1$. For example, for $S^5$, we may write the quasinormal mode frequencies as

$$\omega_*(\Delta_\pm) = \pm\frac{i}{L}(2(\tilde{p}_1 + \tilde{p}_2) + \ell_1 + \ell_2 + \Delta_\pm)\,, \tag{40}$$

We further combine the quantum numbers $2\tilde{p}_i + \ell_i$ into a single integer $k_i$ with degeneracy $k_i + 1$. The four sums over $\tilde{p}_i$ and $\ell_i$ reduce to two sums over $k_1$ and $k_2$, which can be reduced to a single sum over an integer $r$ with degeneracy $d^{S^5}_r = (r+2)^2(r+3)(r+1)/12$.

---

[11]Recall the Hurwitz zeta function

$$\zeta(z,x) = \sum_{r=0}^{\infty}(x+r)^{-z}\,, \quad \partial_z\zeta(z,x) = \zeta'(z,x) = -\sum_{r=0}^{\infty}(x+r)^{-z}\log(x+r)$$

is a meromorphic function in $z$ and $\mathrm{Re}(x) > -1$, with a simple pole at $z=1$.

[12]At first glance this agreement might be surprising given we have neglected the $\mathrm{Poly}(\Delta_\pm)$, which we would expect to contribute. It turns out, however, that the 1-loop partition function (36) is exact because for odd-dimensional spheres $\mathrm{Poly}(\Delta_\pm) = 0$, as is the case for scalar fields [15].

The quasinormal mode frequencies (40) are the Wick rotated normal modes of a scalar field on AdS$_5$ [3]. The introduction of multiple integers $\tilde{p}_i$ and $\ell_i$ is in fact well motivated. Generally, thermal AdS$_{2n+1} \simeq \mathbb{H}^{2n+1}/\mathbb{Z}$, has an associated Selberg zeta function $Z_{\mathbb{Z}}$,

$$Z_{\mathbb{Z}}(z) = \prod_{k_1,k_2,\ldots,k_n=0}^{\infty} \left( 1 - e^{2ib_1k_1} e^{-2ib_1k_2} \ldots e^{2ib_nk_{2n-1}} e^{-2ib_nk_{2n}} e^{-2a(k_1+\ldots+k_{2n}+z)} \right), \tag{41}$$

where $2a$ is the length of the primitive closed geodesic and $e^{2ib_i}$ are the eigenvalues of a rotation matrix describing the rotation of nearby closed geodesics under the Poincaré recurrence map [32]. For thermal AdS$_3$, the integers $k_1$ and $k_2$ can be relabeled such that the zeros of the Selberg zeta function (41) are identified with the poles of the scattering operator [33]. We extended this relabeling to higher AdS$_{2n+1}$ in [3], where we showed it is natural to introduce additional radial and angular momentum integers $\tilde{p}_i$ and $\ell_i$. While there is formally no Selberg zeta function for Euclidean de Sitter space or its quotients, it is interesting that the Wick rotated quasinormal mode frequencies found using the zeros of (41) can be used to build the 1-loop partition function in de Sitter space.

**Lens Space Quotients**

Let us now briefly comment on the computation of 1-loop partition functions for spin-$s$ fields living on $S^{2n+1}/\mathbb{Z}_p$. As in the case of the 1-loop partition function on $S^3$ (24), we can solve for the Euclidean zero modes $\phi_p$, demanding $\phi_p$ satisfy thermal boundary conditions, such that $\Delta_p$ is a negative integer bounded above by the spectrum of the Lens space $S^3/\mathbb{Z}_p$ [34] (rather than the spectrum of the sphere $S^3$). Performing this analysis leads to the 1-loop partition function of a spin-$s$ field on $S^3/\mathbb{Z}_p$

$$\log Z^{(1)}(\Delta_{\pm}) = -\sum_{\pm} \sum_{r=0}^{\infty} d^{(s)}(r) \log(r+s+\Delta_{\pm}), \tag{42}$$

where $d^{(s)}(r)$ is the degeneracy formula for total angular momentum $r$ on $S^3/\mathbb{Z}_p$ [10]

$$d_r^{(s)} = \frac{\tau_2}{2\pi} \left( 1 - \frac{\delta_{s,0}}{2} \right) \sum_{k \in \mathbb{Z}_p} \left[ \chi_{(\frac{r}{2})}(k\tau) \chi_{(\frac{r}{2}+s)}(k\bar{\tau}) + \chi_{(\frac{r}{2}+s)}(k\tau) \chi_{(\frac{r}{2})}(k\bar{\tau}) \right], \tag{43}$$

where $\tau$ and $\bar{\tau}$ are the parameters introduced in (19), and $\chi_{(\ell)}(\tau)$ is the $SU(2)$ character in the representation $\ell$

$$\chi_{(\ell)}(\tau) = \text{tr}_{(\ell)} e^{\frac{i\tau}{2}\sigma_3} = \frac{\sin\left[\frac{(2\ell+1)\tau}{2}\right]}{\sin\left(\frac{\tau}{2}\right)}. \tag{44}$$

The 1-loop partition function 42 can similarly be regulated with the aid of Hurwitz zeta functions, c.f., [5], or the Barnes zeta function [26].

The Euclidean zero mode analysis can be extended to higher dimensional Lens spaces $S^{2n+1}/\mathbb{Z}_p$ in a straightforward way, such that the 1-loop partition function becomes

$$\log Z^{(1)}(\Delta_{\pm}) = -\text{Vol}(S^{2n+1}/\mathbb{Z}_p) \sum_{\pm} \sum_{r=0}^{\infty} \sum_{k \in \mathbb{Z}_p} \chi_r(\gamma^k) \log(r+s+\Delta_{\pm}), \tag{45}$$

where $\chi_r(\gamma^k)$ is the group character of $SO(2n+2)$ in representation $r$, and $\text{Vol}(S^{2n+1}/\mathbb{Z}_p)$ is the volume of the quotient $S^{2n+1}/\mathbb{Z}_p$, e.g., when all of the angular potentials are turned off, $\text{Vol}(S^{2n+1}/\mathbb{Z}_p) = \beta/2\pi$ [11].

It would be interesting to derive the 1-loop partition functions (42) and (45) using the quasinormal mode frequencies $\omega_*(\Delta_\pm)$. Doing so requires knowledge of the Matsubara frequencies $\omega_n(T)$ and the quasinormal mode frequencies $\omega_*(\Delta_\pm)$. As in the case of thermal $AdS_{2n+1}$, the Matsubara frequencies[13] are

$$\omega_{\tilde{n}}(T) = 2\pi i \tilde{n} T - i \sum_{i=1}^{n} \ell_i \vartheta_i T\,, \tag{46}$$

where $\tilde{n}$ is the thermal integer and there is an angular momentum quantum number $\ell_i$ of degeneracy $S^1/\mathbb{Z}_p$ for each potential $\vartheta_i$.

To proceed with the calculation, however, we must know the explicit quasinormal mode frequencies $\omega_*(\Delta_\pm)$ of the Lorentzian continuation of Lens space quotients $S^{2n+1}/\mathbb{Z}_p$. Thus far these quasinormal mode frequencies are not yet known[14], nor can they be acquired from the normal mode frequencies of spin fields on an AdS background using $L_{AdS} = i L_{dS}$. This is because the quasinormal modes of de Sitter space Wick rotate to the quasinormal modes of AdS. Unlike AdS with angular potentials, de Sitter space with angular potentials will have slightly different quasinormal modes, just like for the rotating vs non-rotating BTZ black hole. We will leave this analysis for future study.

## 3.2 Comparing to the Heat Kernel Method

Above we derived the 1-loop partition function for spin-$s$ fields on $S^{2n+1}$ and $S^{2n+1}/\mathbb{Z}_p$ by summing over the Lorentzian $dS_{2n+1}$ quasinormal frequencies, found by Wick rotating the normal frequencies of spin-$s$ fields living on thermal $AdS_{2n+1}$. Alternatively, the 1-loop partition function can be computed using the heat kernel method [10, 11]. In this method the heat kernel $K(x, y; t)$ between two spacetime points $x, y$ is constructed in terms of the eigenfunctions $\psi_r^{(s)}(x)$ and eigenvalues $E_r^{(s)}$ of $\nabla_{(s)}^2$,

$$K^{(s)}(x, y; t) = \sum_r \psi^{(s)}(x) \psi_r^{(s)}(y)^* e^{E_r^{(s)}}\,, \tag{47}$$

where the eigenvalues have degeneracy $d^{(s)}(r)$. The 1-loop partition function for spin-$s$ STT tensor fields on the sphere (or its Lens space quotients) is then computed from the 1-loop determinant

$$\begin{aligned} Z_{(s)}^{(1)} &\propto -\log \det(-\nabla_{(s)}^2 + m_s^2) = -\frac{d}{dz}\left(\frac{1}{\Gamma(z)} \int_0^\infty t^{z-1} K^{(s)}(t) e^{-m_s^2 t}\right)_{z=0} \\ &= \sum_{r=0}^\infty d^{(s)}(r) \log(-E_r^{(s)} + m_s^2)\,, \end{aligned} \tag{48}$$

---

[13] Since $T$ and $\theta$ take on discrete values from identification (19), the $\omega_{\tilde{n}}$ associated with Lens spaces $S^{2n+1}/\mathbb{Z}_p$ depend on the temperature $T$ and a collection of angular potentials $\vartheta_i$. We see that the Matsubara frequencies are also discrete in $p$ and $q$, and reduce to the Matsubara frequencies on the sphere $\omega_n = 2i\pi n$ for $p = 1, q = 0$.

[14] This situation reminds us of the quasinormal mode analysis for fields on a rotating BTZ black hole. The rotation $J$ of the black hole modifies the Lorentzian quasinormal mode frequencies (of a non-rotating black hole), where $J$ is also found to be proportional to the angular potential of the Euclidean theory [18, 20]. Therefore, the rotating BTZ black hole is the Lorentzian continuation of $\mathbb{H}^3/\mathbb{Z}$, where regularity demands we periodically identify the Euclidean time circle and include angular potentials. This is contrary to thermal AdS, where first we Euclideanize AdS and then, by our own will, periodically identify the Euclidean time circle and angular coordinates, introducing an arbitrary temperature and angular potentials. Since, as in the black hole case, the temperature and angular potentials of the Lens space quotients are imposed on us to maintain regularity of the Euclidean solution, the addition of angular potentials are expected to modify the quasinormal mode frequencies of de Sitter space.

where $K^{(s)}(t)$ is the trace of the coincident heat kernel

$$K^{(s)}(t) \equiv \int d^{2n+1}x \sqrt{-g} K^{(s)}(x,x;t) = \sum_r d^{(s)}(r) e^{E_r^{(s)} t} \,. \tag{49}$$

For spin-$s$ fields on $S^{2n+1}$, the eigenvalues are given by

$$E_r^{(s)} = -(r+s+n)^2 + s + n^2 \,, \tag{50}$$

with degeneracy $d^{(s)}(r)$ is given by (32). For spin-$s$ fields on $S^{2n+1}/\mathbb{Z}_p$, the heat kernel is built using the method of images (4), where the eigenvalues are the same as in (50) but with degeneracy $d^{(s)}(r) = \mathrm{Vol}(S^{2n+1}/\mathbb{Z}_p) \sum_{k \in \mathbb{Z}_p} \chi_r(\gamma_p^k)$, e.g., (43) for $S^3/\mathbb{Z}_p$.

Let us now compare the quasinormal mode and heat kernel methods. Earlier we saw that summing over the quasinormal frequencies builds the degeneracy $d^{(s)}(r)$. In the heat kernel method the degeneracy $d^{(s)}(r)$ corresponds to the degeneracy of the eigenvalues $E_r^{(s)}$ of $\nabla^2_{(s)}$. The quasinormal mode method too carries this information. Expanding the sum over the conformal dimensions $\Delta_\pm$ in (31) or (42)

$$
\begin{aligned}
\sum_\pm \log(r+s+\Delta_\pm) &= \log(r+s+n+\sqrt{s+n^2-m_s^2}) + \log(r+s+n-\sqrt{s+n^2-m_s^2}) \\
&= \log[(r+s+n)^2 - s - n^2 + m_s^2] \\
&= \log(-E_r^{(s)} + m_s^2) \,.
\end{aligned}
\tag{51}
$$

Using this relation we find exact agreement between the 1-loop partition functions computed using quasinormal mode and heat kernel methods. Altogether we see that the sum over the conformal dimension $\Delta_\pm$ encodes the eigenvalues $E_r^{(s)}$ of Laplacians $\nabla^2_{(s)}$ on $S^{2n+1}$, while the sum over the quantum numbers $\tilde{p}$ and $\ell$ of the quasinormal mode frequencies, whatever they end up being, builds the degeneracy $d^{(s)}(r)$ of the eigenvalues $E_r^{(s)}$, similar to the case of thermal AdS [1–3].

## 4 Discussion

In this note we computed the 1-loop partition function of spin-$s$ fields on Euclidean de Sitter space $S^{2n+1}$ using the quasinormal mode method. Rather than computing the quasinormal modes from scratch, we instead used the analytic continuation prescription $L_{\mathrm{AdS}} \to i L_{\mathrm{dS}}$ appearing in the dS/CFT correspondence, and used the Wick rotated normal frequencies of thermal AdS$_{2n+1}$. We showed how the Euclidean zero mode analysis used in the Euclidean de Sitter space readily generalizes to Lens spaces $S^{2n+1}/\mathbb{Z}_p$, and commented on how the problem can be tackled using the quasinormal mode frequencies. We then compared the quasinormal mode method of calculating 1-loop determinants to the heat kernel method, where we found that the sum over conformal dimensions $\Delta_\pm$ carries all of the information of the eigenvalues $E_r^{(s)}$ of the spin-$s$ Laplacian $\nabla^2_{(s)}$, and the sum over the quasinormal frequency integers $\tilde{p}$ and $\ell$ build the degeneracy $d_r^{(s)}$ of the eigenvalues. Our analysis shows that the quasinormal frequencies encode the group structure of the space the fields live on, akin to observations made for thermal AdS [1–3]. Interestingly, despite the fact Eucliden de Sitter space does not have a Selberg zeta function, we observed that the Lorentzian quasinormal mode frequencies for de Sitter space are the Wick rotated normal mode frequencies of AdS, whose Euclidean zero modes can be found from the zeros of the Selberg zeta function associated with hyperbolic quotients.

There are some possible extensions and applications our work which might be of interest. First, it would be worthwhile to directly compute the quasinormal modes for spin-$s$ fields on

$S^{2n+1}/\mathbb{Z}_p$, along the lines of [27,35], as this would provide further evidence to the relationship between AdS/CFT and dS/CFT correspondences. One application of our work includes studying higher spin bosonic contributions to the Hartle-Hawking wavefunction to see whether the higher spins help in regularizing the full partition function, similar to what was shown for the three-dimensional case in [36]. Moreover, in three-dimensions our analysis readily extends to spinor fields, and it therefore would be possible to study 1-loop partition functions in de Sitter supergravity. Another potential application would be to study quantum corrections to entanglement entropy in de Sitter space [37] analogous to the thermal AdS case [38], which would require computing 1-loop partition functions on more general quotients $S^{2n+1}/\Gamma$.

# Acknowledgements

We would like to thank Cynthia Keeler for helpful discussions. The work of VM is supported by the U.S. Department of Energy under grant number DE-SC0018330.

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
