# Peer review of "Higher spin partition functions via the quasinormal mode method in de Sitter quantum gravity"

_SciPost Physics, doi:SciPost Phys. 9, 039 (2020)_

## Round 1 · Referee Report · Anonymous (Referee 1) · 2020-5-22

Strengths

1) The paper is concise

2) Reproduces a previous calculation while using a streamlined method

Weaknesses

1) Somewhat confusing use of terminology (see report)

Report

This paper nicely reproduces the partition function for fields of arbitrary spin on the Euclidean sphere $S^N$ and its Lens-space quotients, using the quasinormal mode method. This has the interpretation of a thermal partition function in the static patch.

This paper should be published, however I have two minor comments/confusions:

1) I am puzzled by the use of the word "Dirichlet boundary condition" in the paper. The Euclidean continuation of de Sitter is compact, so there is nowhere to place a Dirichlet condition. As far as I can tell from the paper, the term is used to mean `periodic in Euclidean time.' If I am correct about this interpretation, I think the authors should change this where appropriate.

2) Right before section 3.2 the authors comment that they do not know the quasinormal modes of the Lorentzian Lens spaces. But as far as I'm aware, the zero-mode solutions they compute in Euclidean respecting thermal boundary conditions can be used to define these Lorentzian quasinormal modes. If possible, I would the authors to clarify these remarks somewhat.

Minor typo:

Missing section reference at the end of the introduction

Requested changes

1) Clarify or remove the use of Dirichlet boundary conditions

2) Clarify the discussion around Lens-space QNMs

  • validity: high
  • significance: good
  • originality: good
  • clarity: ok
  • formatting: excellent
  • grammar: perfect

Author:  Andrew Svesko  on 2020-07-03  [id 872]

(in reply to Report 1 on 2020-05-22)

We thank the referee for their constructive and thought-provoking feedback. Below we will address their questions and comments.

The referee brings up two points: (i) the use of the phrase Dirichlet boundary conditons, and (ii) the existence of Lorentzian quasinormal modes for Lens space quotients. The first of these points the referee rightly recognizes that by Dirichlet boundary conditions we really mean boundary conditions which respect the thermality condition that Euclidean time be periodic; indeed it does not make sense to refer to Dirichlet boundary conditions for fields over a compact sphere. As such, we have removed this phrase throughout the paper and replaced it with boundary conditions periodic in Euclidean time, which we also sometimes refer to as thermal boundary conditions.

The second point is more interesting, which we have addressed in footnote 13. For comparison, let us consider the analogous analysis for fields on a rotating BTZ black hole background. The rotation $J$ of the black hole modifies the Lorentzian quasinormal mode frequencies (of a non-rotating black hole), where $J$ is also found to be proportional to the angular potential of the Euclidean theory, as recognized in Refs. [18] and [20]. Therefore, the rotating BTZ black hole is the Lorentzian continuation of $\mathbb{H}^{3}/\mathbb{Z}$, where regularity demands we periodicially identify the Euclidean time circle and include angular potentials. This is contrary to thermal AdS, where first we Euclideanize AdS and then, by our own will, periodically identify the Euclidean time circle and angular coordinates, introducing an arbitrary temperature and angular potentials. Since, as in the black hole case, the temperature and angular potentials of the Lens space quotients are imposed on us to maintain regularity of the Euclidean solution, the addition of angular potentials are expected to modify the quasinormal mode frequencies of de Sitter space. In short, we expect that the angular potentials appearing in Matsubara frequencies will also appear in the quasinormal mode frequencies of the Lorentzian continued Lens space quotients -- such quasinormal mode frequencies are not yet known.

We hope that these clarifications will adequately satisfy the referee's request, and believe that commenting on these points in the article as we have done improves the original version of this article.

---

## Round 1 · Referee Report · Anonymous (Referee 2) · 2020-5-23

Strengths

1- The computations done seem correct. 2- The authors show that the quasinormal mode method to evaluate determinants on $S^{2n+1}$ agrees with known results using other methods.

Weaknesses

1- The analysis is incomplete. 2- Scope and impact of the results here is minimal. 3- In higher dimensional de-Sitter spaces (d>3), other topologies could also contribute to the Euclidean path integral. One example would be euclidean black holes in dS. Those spaces are not mentioned nor acknowledged in this analysis.

Report

This manuscript fills a minor technical gap to show that different methods to evaluate one-loop determinants agree. My reason to reject this manuscript is two-fold:

1- The advances presented in this work are very limited. One motivation of the authors is to further explore the relation between AdS/CFT and dS/CFT, and they confirm that certain results can be analytically continued from AdS to dS. The verifications that different methods agree and the analytic continuation works, while reassuring, are not surprising nor provide new insights. 2- I think this manuscript does not have enough new results, and certain portions are incomplete. For example, the analysis of lens spaces is superficial: the subsubsection at the end of section 3.1 contains no new results, just a summary of other work.

There are also a few obvious typos: the last paragraph of page 3 has a reference missing; the sentence below eqn (3.7) is incomplete.

  • validity: ok
  • significance: ok
  • originality: ok
  • clarity: good
  • formatting: reasonable
  • grammar: good

Author:  Andrew Svesko  on 2020-07-10  [id 879]

(in reply to Report 2 on 2020-05-23)
Category:
remark

We thank the referee for taking the time to provide us with feedback. The referee has two central critiques: (i) the impact of the results is minimal and (ii) the analysis is incomplete. We will briefly respond to these comments in turn.

More specifically in point (i) they argue a motivation of our work was to further study the relation between AdS/CFT and dS/CFT, for which our paper merely confirms though does not provide any surprising or new insights. We agree a motivation of our work was to compare these correspondences, however, to do so via the analysis of quasinormal modes, by analytically continuing the normal modes of AdS, which has only been discussed very briefly in the literature for specific types of fields. That is, our goal was to exploit the prescription $L_{AdS}\to iL_{dS}$ and compute functional determinants on de Sitter space and its Lens space quotients, and see whether our resulting expression agrees with the established heat kernel result. Here we generalized this realization for higher spin fields, which we believe is an important check, and may have further use in the future.

In point (ii) the referee argues that the analysis is incomplete, in particular pointing to the content devoted to Lens spaces. As discussed in the article, we show how the functional determinants for higher spin fields on higher dimensional Lens spaces would arise from solving the Euclidean zero modes, and verify the determinants match the heat kernel result. We further discussed why we could not complete the analysis with via quasinormal frequencies (they are not known at this time), and, in effect, point to a limit of the prescription $L_{AdS}\to iL_{dS}$. This question is interesting of course, and is a complex enough problem, that we are saving it for a separate future article.

The referee also points out a couple of typos, which we have fixed, and that a weakness of our article is not mentioning the fact that other topologies contribute to the partition function in higher dimensional de Sitter spaces. There are in fact other kinds of topologies that contribute to the Euclidean partition function for three dimensional de Sitter space, which we do mention briefly. However we point out that the goal of this work was not to compute the entire partition function for higher dimensional de Sitter space, or the functional determinants of fields on all possible topologies which might contribute to the partition function. Rather we were simply trying to calculate the functional determinants of arbitrary spin fields on specific topologies that contribute to the full partition function.

All in all, we believe our paper may be brief, though not incomplete and not minimal. We thank the referee again for their thoughts and feedback.

---

## Round 1 · Referee Report · Anonymous (Referee 3) · 2020-6-25

Strengths

  1. This paper is well written.

Weaknesses

  1. This is a rather straightforward application of the quasinormal mode techniques to calculate the partition function.

Report

This paper evaluates one-loop determinants using the quasinormal mode method for $dS_{2n+1}$. It generalises the analysis of reference [15] to the case of spin-$s$ fields and finds agreement with the heat kernel method.

Although the authors have ignored the ${\rm Poly}(\Delta)$ factor they find agreement with previous results in the literature. In reference [15] this factor was shown to be trivial only for the case of scalars for odd-dimensional spheres. Can the authors provide a justification why this should hold true for the case of higher spin fields as well?

The authors mention that the ${\rm Poly}(\Delta)$ factor doesn't contain temperature dependence. However, the final result for the one-loop determinant turns out to be a constant (for instance, in equation (3.20)) unlike AdS where an explicit temperature (modular parameter) dependence is present. This being the case, it might be a priori difficult to separate regularized pieces from ${\rm Poly}(\Delta)$ and the rest of the expression for $Z^{(1)}$. In view of this, I request the authors to provide an explanation for why it can be ignored or why the factor is trivial.

Requested changes

  1. I request the authors to address the question I have raised above.

  2. Can the authors comment on what are the potential obstacles to addressing the problem in $dS_{2n}$ using the QNM approach? This situation can be interesting in the context of higher spin holography for $dS_4$ (although computations for the 1-loop partitions function by heat kernel methods already exist).

  • validity: ok
  • significance: ok
  • originality: ok
  • clarity: good
  • formatting: good
  • grammar: perfect

Author:  Andrew Svesko  on 2020-07-03  [id 871]

(in reply to Report 3 on 2020-06-25)
Category:
answer to question

We thank the referee for their constructive and thought-provoking feedback. Below we will address their questions and comments.

The referee brings up two points: (i) why the $\text{Poly}(\Delta)$ term can be ignored, or why it is trivial, and (ii) commenting on the potential obstacles to addressing the problem in even dimensional de Sitter space.

Let us respond to (i) first. From the quasinormal mode perspective, technically we should include $\text{Poyl}(\Delta_{\pm})$ throughout our calculations the entire time. To determine the full 1-loop partition function, then, we would need to uncover the form of $\text{Poly}(\Delta)$. This cannot be accomplished by the quasinormal mode method by itself; instead we fix $\text{Poly}(\Delta)$ by requiring the correct large $\Delta$ behavior found, e.g., via the heat kernel coefficients of the Laplacian in the large $\Delta$ limit. In other words, it is only after we compare the quasinormal method to an alternative method in the large $\Delta$ limit that we find $\text{Poly}(\Delta)=0$ for odd-dimensional spheres. This is what Ref. [15] finds in the case of scalar fields.For higher spin fields, a priori we should keep the $\text{Poly}(\Delta)$ contribution throughout all of our calculations. However, upon comparing to the heat kernel approach, we find that in the case of odd-dimensional spheres the quasinormal mode and heat kernel methods match when $\text{Poly}(\Delta)=0$. We have clarified this in the pargraph before Eq. (3.7). Lastly, we note that in our calculation of the 1-loop partition function we set $T=1/2\pi$. Had we kept $T$ explicitly, it is manifest that $\text{Poly}(\Delta)$ does not have temperature dependence.

Now we address point (ii). There is mainly one potential obstacle with completing the quasinormal mode analysis on even dimensional spheres. As demonstrated in Ref. [15] -- specifically for scalar fields living on even-dimensional spheres, though similar arguments hold for higher spin fields -- $\text{Poly}(\Delta)$ does not vanish in even dimensions, unlike their odd-dimensional counterparts. In fact, it will depend on the regularization scale. Consequently, one must include the $\text{Poly}(\Delta)$ contribution and determine its form upon comparing to its heat kernel counterpart in the large $\Delta$ limit. This was shown explicitly for scalar fields in particular dimensions in Ref. [15], however, has not been shown for higher spin fields. Secondly, while not a chief obstacle, in the article we worked with writing the geometry of the sphere $S^{2n+1}$ in terms of Hopf-like coordinates, where it is easy to identify the angular potentials $\phi_{i}$. For even-dimensional spheres $S^{2n}$ one instead works with spherical coordinate geometry, where we would define the angular potentials differently. Moreover, we would need additional information about even-dimensional Lens spaces, which are less commonly discussed in the literature and whose spectra are unknown to us.

We hope that these changes and clarifications adequately satisfy the referee's request, and believe the referee's requested changes improves the original version of this article.

---

## Round 2 · List of Changes

-- Rename Dirichlet boundary conditions to boundary conditions periodic in Euclidean time
-- Typos fixed

---

## Editorial Decision

published